# Can a specific biobehavioral-based therapeutic education program lead to changes in pain perception and brain plasticity biomarkers in chronic pain patients? A study protocol for a randomized clinical trial

Silvia Di Bonaventura[1,2,3], Josué Fernández Carnero[2,3,4,5]*, Raúl Ferrer-Peña[3,4,5,6]

1 Escuela Internacional de Doctorado, Department of Physical Therapy, Occupational Therapy, Rehabilitation and Physical Medicine, Rey Juan Carlos University, Alcorcón, Spain, 2 Department of Physical Therapy, Occupational Therapy, Rehabilitation and Physical Medicine, Rey Juan Carlos University, Alcorcón, Spain, 3 Cognitive Neuroscience, Pain and Rehabilitation Research Group (NECODOR), Faculty of Health Sciences, Rey Juan Carlos University, Madrid, Spain, 4 La Paz Hospital Institute for Health Research, IdiPAZ, Madrid, Spain, 5 Motion in Brains Research Group, Centro Superior de Estudios Universitarios La Salle, Universidad Autonóma de Madrid, Madrid, Spain, 6 Departamento de Fisioterapia, Facultad de Ciencias de la Salud, CSEU La Salle, Universidad Autonóma de Madrid, Madrid, Spain

* josue.fernandez@urjc.es

**Data Availability Statement:** No datasets were generated or analysed during the current study. All

## Abstract

### Background

Chronic pain conditions are complex multifactorial disorders with physical, psychological, and environmental factors contributing to their onset and persistence. Among these conditions, the role of brain-derived neurotrophic factor (BDNF) and the impact of a specific therapeutic education (TE) on pain management have emerged as important areas of research.

### Objective

This study aims to investigate the effects of a specific type of therapeutic education on pain levels and BDNF concentrations.

### Methods

In this single-blind, randomized clinical trial, patients will be randomly assigned to one of two groups: one will receive exercise with TE and the other without TE. Assessments will be made at baseline, mid-treatment, post-intervention, and at one and eight months.

### Outcomes

This study will shed light on the effectiveness of a therapeutic education (TE) program in pain management. Additionally, it will provide information on its effects on BDNF levels, a biomarker of brain plasticity, as well as on various psychosocial variables that can influence pain experience.

relevant data from this study will be made available upon study completion.

**Funding:** The authors received financial support for the future randomized clinical trial from the CSEU La SALLE (Universidad Autónoma de Madrid) with grant number: 2022A36005 but not for this protocol. The authors who have received funding are not part of the present study as it is a shared project.

**Competing interests:** The authors have declared that no competing interests exist.

## Conclusion

By comprehensively addressing the need to quantify brain changes more precisely in individuals with chronic pain during interventions like TE and recognizing the importance of establishing a more structured and comprehensive protocol, this study lays a solid and replicable foundation for future evidence-based treatment developments.

## Background

Chronic pain is a widely prevalent health problem that affects one in five people worldwide [1]. It is a debilitating condition with a potentially significant impact on one's quality of life, affecting one's ability to perform daily activities and decreasing one's emotional and physical well-being [2], as well as the resources needed to cope therewith [3]. One serious drawback that accompanies chronic pain is the frequent dependence on drugs for pain relief. In particular, opioids have become one of the main medications for treating pain, as also one of the causes of the increase in mortality due to overdose [4] However, physiotherapy modalities that include therapeutic education and exercise are a common non-pharmacological therapeutic approach for chronic pain and its maladaptive plasticity [5–7], defined as a change in the structure or function of the nervous system resulting in increased pain sensitization [8, 9]. Specifically, therapeutic education (TE) plays a crucial role in reversing maladaptive plasticity, reducing pain and disability, fear avoidance, and pain catastrophizing, *inter alia* [10] TE is a fundamental component of chronic pain management and addresses several aspects essential for improving patients' quality of life [11, 12]. These include understanding pain and its mechanism of production [13], promoting self-regulation and self-care strategies [14], acquiring coping skills [15], reducing fear-avoidance and pain catastrophizing [16], promoting healthy lifestyles, and optimizing physical functionality [17]. One of the main goals for the patient is to reconfigure maladaptive beliefs associated with their chronic pain, involving a change in the structure and function of the nervous system that has contributed to increased pain sensitization [18, 19].

However, no random educational approach is appropriate for this. There are several types of TE, addressing different topics [20, 21] and using different dosages [22–24], which inevitably lead to different results [25–27]. A specific approach is required that considers both the biological and behavioral aspects of chronic pain. Further, to accurately determine the true impact of such non-only-informative intervention, it is imperative to measure and evaluate the effects of the TE not only in terms of perceived pain and psychosocial factors but also through the lens of brain plasticity biomarkers and draw correlations within these domains.

In this sense, biological components like brain-derived neurotrophic factor (BDNF) and its relationship to cerebral plasticity in patients with persistent pain [28, 29]. BDNF, a protein that plays an important role in pain sensitization and neuronal plasticity, has been associated with neuronal hyperexcitability in chronic pain [30]. Understanding its relationship to pain is crucial for developing new therapies. Manipulating BDNF expression could offer options for modulating neuronal plasticity and relieving pain in patients with chronic pain. However, the impact of these interventions on the adaptive changes in the brain, which are crucial in patients with persistent pain, remains largely unknown. Additionally, a lack of adherence to established guidelines for reporting interventions, such as the TIDieR checklist for general interventions [31], or GREET checklist for educational interventions [32] and the Consensus on Exercise Reporting Template (CERT) checklist for exercise interventions [33] hinders their replication and adaptation to individual patients, thus limiting their practical utility [34].

In summary, despite the challenges in the field of education and exercise for chronic pain management, this model of pain TE will address the aforesaid needs, providing a comprehensive and evidence-based approach that considers changes in brain plasticity learning-related, biobehavioral aspects, appropriate dosing, and objective evaluation of results.

## Materials and methods

### Study setting

The proposed study is a single-blind, parallel-group, randomized control trial. Its protocol follows the structure based on the PICO question (Population, Intervention, Comparison, Outcome) [35] and adheres to the recommendations of the Standard Protocol Items: Recommendations for Interventional Trials (SPIRIT) guideline [36]. In addition, specific interventions were described based on the Template for Intervention Description and Replication (TIDieR) checklist [31], the Guideline for Reporting Evidence-based Practice Educational Interventions and Teaching (GREET) [32], and the Consensus on Exercise Reporting Template (CERT) checklist [33]. The trial was approved by the ethics committee of Rey Juan Carlos University (Ethical number: 1901202202822) and the entire study will be conducted at the Universidad Rey Juan Carlos located in Alcorcón, Madrid (Spain). The trial identifier code is NCT05623579, registered on November 18, 2022, in ClinicalTrials.gov.

### Eligibility criteria

Patients will be recruited from the physical therapy clinics located in Madrid, Spain, based on the following inclusion criteria: aged between 18 and 65, having a history of pain lasting more than twelve weeks, pain corresponding to at least 3 out of 10 on the visual analog scale, possessing the ability to comprehend, speak, and write in Spanish, not currently undergoing any other treatments for 3 months before the study, and explicitly consenting to not being allowed to initiate new treatments till the end of the follow-up period. The exclusion criteria are neurological cognitive alteration that prevents understanding the contents of the educational program (in case of doubt, assessment with the Mini-Mental test will be conducted, requiring a minimum score of 25), systemic, oncological, or inflammatory diseases, psychiatric pathologies, pregnancy, diabetes type II and inability to adhere to the protocol.

The following professionals were selected to conduct the study: two physiotherapists with more than 15 years of experience in the treatment of chronic pain for the pain education and exercise therapy component, two nurses for blood collection, another physiotherapist for administering questionnaires on psychosocial variables and quantitative sensory testing, and two biochemists for BDNF analysis.

### Interventions

All patients who meet the inclusion criteria and agree to participate in the study will be informed verbally and in writing, with the delivery of the information sheet listing the specific characteristics of the study. They will be asked to sign the written informed consent form.

Sociodemographic data will then be collected, as well as psychosocial variables on their current pain process. This data collection will be carried out in a room in the physiotherapy gymnasium of the university center, previously designated for measurements by the evaluating researcher as well as blood extraction.

Once the specific questionnaires and the pressure pain threshold (PPT) at the joints of the elbow and tibialis anterior are measured, a trained professional will collect 5 ml of blood from the ulnar vein of the participants in a citrate vacutainer. The blood collection will always take

place between 6 p.m. and 7 p.m., to minimize the possible effects of circadian changes [37–39]. Blood samples will be collected in tubes containing K3-EDTA and centrifuged at 1000 g for 10 min at 4°C. The supernatants for the total plasma (TP) condition will be frozen at -80°C. Blood collection will be performed in both groups at rest before the intervention, 15 days after the intervention, at the end of the intervention period, and at 1 month and 8 months after the intervention.

After the initial data collection, a researcher other than the initial evaluator, and the physiotherapist-therapist who will apply the subsequent treatment to ensure blinding, will proceed to randomly assign the subjects to one of the two study groups through a list previously generated with GraphPad software (version 5.01 Graphpad software, Inc. San Diego, California, USA).

### Intervention group (Exercise + POBTE)

Patients assigned to the intervention group will be instructed by a physiotherapist in the performance of the TE sessions according to their functional capacity, with reference to their HRmax [40]. They will be prescribed the reference intensity measures for the expected increase in training time indications on potential risks and references for the interruption of the exercise in their case. The duration of the exercise program will be 4 weeks, with 3 sessions lasting a maximum of 45 minutes per session. On alternate days, twice a week, the patient will receive *Pain Oriented Biobehavioral Therapeutic Education (POBTE)* sessions, focusing on the specific parameters described below. The duration of the sessions will be a maximum of 45 minutes each day [41], and the education program will be spread across 8 sessions over 4 weeks. The intervention group will be instructed to refrain from discussing treatment details with participants assigned to the control group until the end of the treatment and follow-up periods.

The main learning areas that to be addressed and the biobehavioral interventions proposed are summarized in Fig 1, and the conceptual framework of the POBTE intervention is explained in detail in Fig 2 and S1 File. The specific cognitive components of the intervention are listed in the general intervention checklist, following the TiDier checklist (see S1 Checklist) and the exercise intervention is presented in S2 Checklist with the CERT checklist, while the Therapeutic Educational Intervention is presented in S3 Checklist with the GREET checklist.

A chronological representation of the tasks to be performed during the POBTE intervention is displayed in Fig 2. This figure is arranged to correspond to the stages of the meaningful learning phase in the intervention design.

### Active control group (Exercise)

Patients assigned to the active control group will be instructed by a physiotherapist in the performance of TE sessions according to their functional capacity, with reference to their HRmax. They will be prescribed the reference intensity measures for the expected increase in training time, indications of potential risks, and references for the interruption of the exercise in their case. The duration of the exercise program will be 4 weeks, with 3 sessions lasting a maximum of 45 minutes per session. Contents of the exercise intervention are described in the CERT checklist (S2 Checklist).

### Intervention modifications

The criteria for discontinuing or modifying allocated interventions for a given trial participant would include participant request, adverse events, or harms, improvement in or worsening of the disease, and non-compliance. In the context of this study, criteria will be adapted to the specific needs and characteristics of the study population.

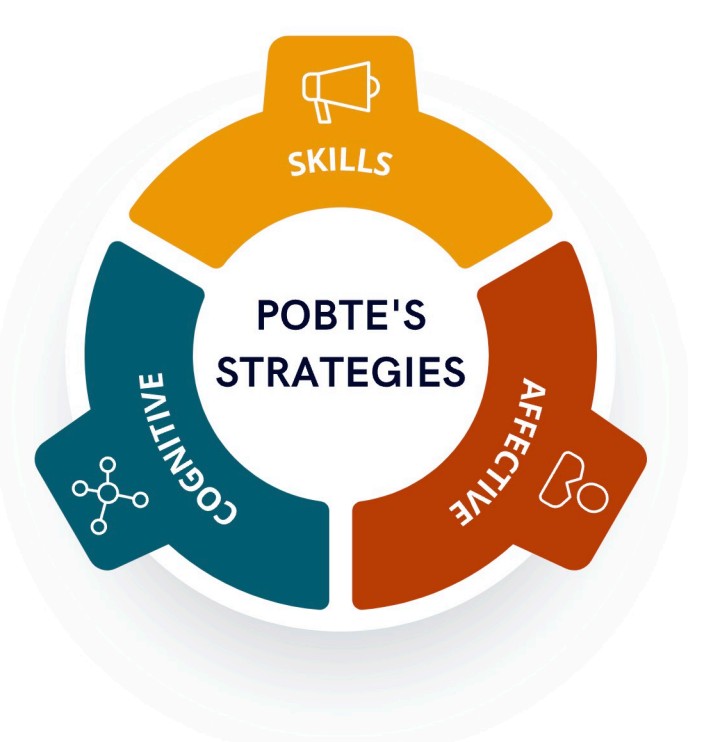

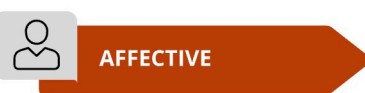

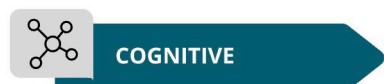

**Fig 1. POBTE's strategies.**

### Intervention adherence

To improve adherence, we will provide clear instructions, personalized feedback, and support, simplify the intervention as much as possible without losing scientific rigor, and monitor attendance, self-reports, and reminders. Written materials and videos will be provided, along with personalized feedback, progress reports, and regular check-ins. Attendance, self-reports, and reminders will be used to monitor adherence.

## Outcomes

### Primary outcome

**Pain intensity.** *Visual Analogue Scale (VAS)*. Pain intensity will be measured using a 100 mm visual analog scale where 0 represents 'no pain' and 100 the 'worst pain imaginable'. Participants draw a mark at a point on the line that best reflects the pain they are experiencing at the time of measurement. Higher scores indicate higher pain levels. The sensitivity and specificity of this questionnaire and the acceptability of its psychometric properties have been approved [42].

### Secondary outcomes

**Biochemical.** *Plasma BDNF levels*. Within 30 minutes of collection, the blood samples will be centrifuged and the plasma separated into 0.5 ml aliquots for further analysis. Plasma BDNF levels will be determined by enzyme-linked immunosorbent assay (ELISA) test using monoclonal antibodies specific to neurotrophin (R&D Systems, Minneapolis, MN), using the

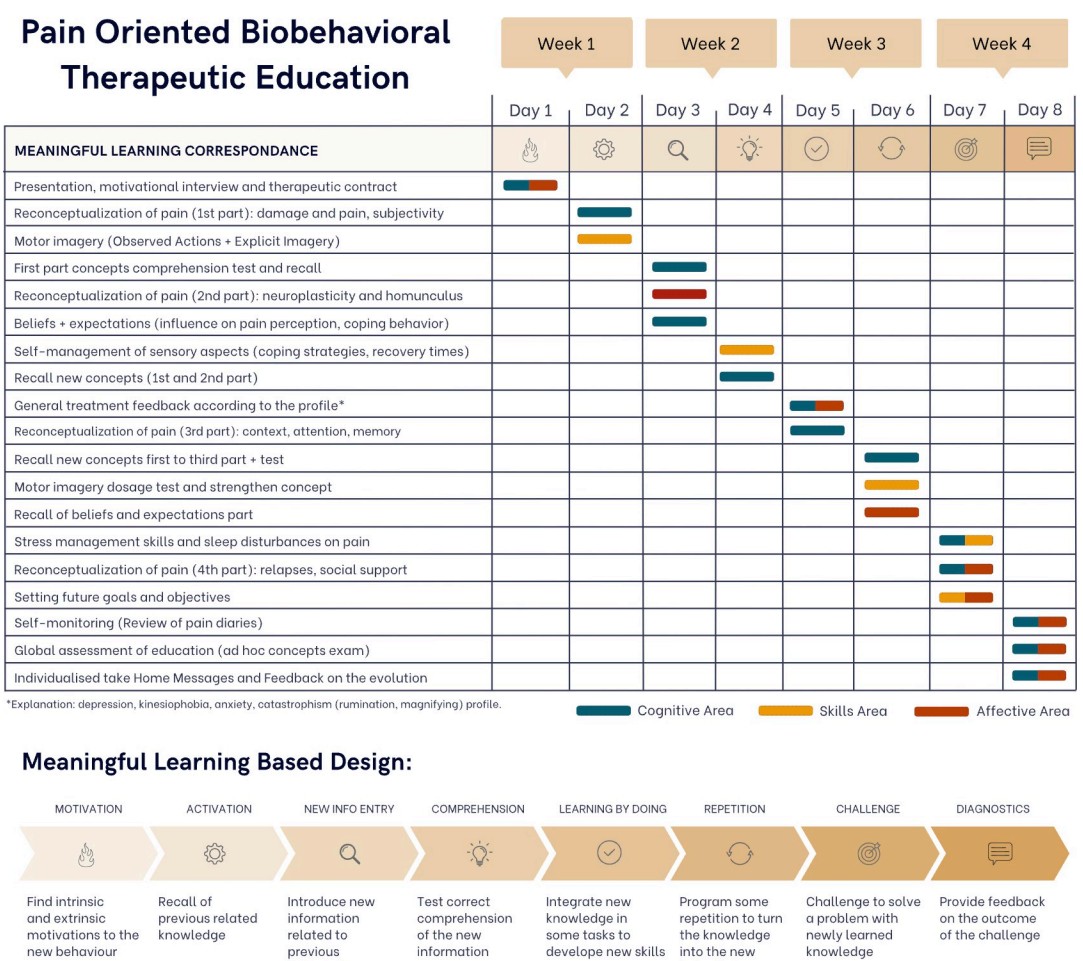

**Fig 2. POBTE's intervention.**

manufacturer's protocol. All samples will be tested in duplicate to avoid intra-assay variations. The lower limit of detection for BDNF from the kit is 7.8 pg/mL. Assay (ELISA) will use a ChemiKine BDNF ELISA kit, CYT306 (Chemicon/Millipore, Billerica, MA). Optical density will be measured using an ELISA reader at a wavelength of 450 nm (GloMax-Multi Microplate Reader, Promega, Madison, WI) for multiplex assay measurements. Data will be expressed in pg/mg protein.

**Psychosocial.** *Anxiety and depression* will be assessed using the Spanish-validated version of the HADS self-completion scale. This scale is divided into two subscales of 7 items each: 1) depression (HADSDep) and 2) anxiety (HADSAnx). The HADS has demonstrated good reliability and validity in various pathologies [43].

*Quality of life.* Measured with the EuroQoL-5D questionnaire (EQ-5D), this is a self-report instrument to assess the health-related quality of life comprised of three items, a 5-factor descriptive scale, a second item comprising a vertical VAS and a third one about social values index generated by the instrument. The EQ-5D has shown good psychometric characteristics [44].

*Catastrophism* will be measured by the Spanish version of the Pain Catastrophism Scale (ECD). This has demonstrated adequate psychometric characteristics for the assessment of this construct with internal validity (Cronbach's alpha 0.81) [45].

*Chronic Pain Grading Scale (CPGS)* is a self-report instrument consisting of an 8-item scale. It has a high internal consistency, with a Cronbach's α of 0.87, similar to that of other language versions, and an intraclass correlation coefficient of 0.81. The average administration time is 2 min 28 s [46].

*Tampa Scale for Kinesiophobia (TSK)* is a self-report questionnaire comprising 17 items. The internal consistency of the TSK is high, with Cronbach's alpha coefficients ranging from 0.74 to 0.93, indicating strong reliability. Test-retest reliability is also good, with correlation coefficients ranging from 0.75 to 0.88 [47].

*Pittsburgh Sleep Quality Index (PSQI)* is a self-rated questionnaire of 19 items categorized into seven components, viz., subjective sleep quality, sleep latency, sleep duration, habitual sleep efficiency, sleep disturbances, use of sleep medication, and daytime dysfunction. The internal consistency of the PSQI is high, with Cronbach's alpha coefficients ranging from 0.77 to 0.83, indicating strong reliability. Test-retest reliability is also good, with correlation coefficients ranging from 0.85 to 0.87 [48].

*Perceived Stress Scale* is a self-report questionnaire comprising 10 items, each rated on a 5-point Likert scale ranging from 0 (never) to 4 (very often). The internal consistency of the PSS is high, with Cronbach's alpha coefficients ranging from 0.78 to 0.91, indicating strong reliability. Test-retest reliability is also good, with correlation coefficients ranging from 0.67 to 0.85 [49].

*International Physical Activity Questionnaire (IPAQ)* is a self-reported questionnaire of 27 items to assess the frequency, duration, and intensity of physical activity. The internal consistency of the IPAQ is moderate to high, with Cronbach's alpha coefficients ranging from 0.73 to 0.95. Test-retest reliability is good, with correlation coefficients ranging from 0.70 to 0.88 [50].

*Chronic Pain Self-Efficacy Scale (CPSS)* is a self-reported questionnaire comprising 22 items rated on a 0–10 scale, with higher scores indicating greater self-efficacy in managing pain. The internal consistency of the CPSS is high, with Cronbach's alpha coefficients ranging from 0.89 to 0.95, indicating strong reliability. Test-retest reliability is also good, with correlation coefficients ranging from 0.63 to 0.90 [51].

*Knowledge questionnaire on specific aspects of pain, ad hoc,* includes 5 questions in multi-choice test format on the contents used in the education sessions.

*Global Perception of Change is* the only one that will be tested after the intervention. This instrument consists of a 100mm line numbered in centimeters from -5 to 5 where the left end reads "Much Worse", 0 is printed as "no change" and 5 is marked as "fully recovered". Relative to these values, the patient is asked to answer the following question: "In relation to your pain, how would you describe your current state of health compared to when your pain began?" This version of the change perception scale has been assessed in the literature as the most appropriate in musculoskeletal pain processes, and as a valid and reliable instrument [52].

**Others.** *Pressure Pain Thresholds (PPT).* Pressure pain thresholds will be assessed through a pressure algometer. The algometer will be applied to different points in the body (elbow and knee) and participants will be asked to indicate the point at which pressure changes to pain. The pressure will be gradually increased at approximately 1 kg/s until the participant indicates feeling pain. The threshold pressure at which the sensation changes from pressure to pain will be recorded in kilograms (kg) and used as an indicator of PPT. Higher PPT values indicate higher pain thresholds. The validity and reliability of using a pressure algometer to measure PPT have been established [53].

*Height and weight.* Both these variables will be determined by patient estimation, with no exploration of them in the measurement process.

*Sociodemographic*. This comprises Age, Marital status (Single, Married, Divorced, Widowed), Employment status (Active, Unemployed, Retired, Temporary disability, Student, Homemaker, Other), Educational level (none, primary, secondary, university), and Gender.

*Exercise*. Among the exercise variables, the parameters described below will be considered:

*Blood pressure and respiratory rate*. Measured with the Anura application, which has demonstrated reliability with a conventional blood pressure measurement of 98% [54].

*Saturation and HR* are Measured with an Oxylink finger pulse oximeter.

All the outcome measurements will be assessed in a random order at baseline, at the halfway point of the treatment, post-intervention, and one- and eight-months post-intervention (Figs 3–5).

**Participants' timeline.** We have included a schematic diagram outlining the time frame for participant enrolment, interventions, assessments, and visits, as shown in the following figures (Figs 4 and 5).

**Sample size.** The determination of the sample size used a t-test for independent means, calculated using G*Power software (Universität Düsseldorf, Germany). The significance level was set at 0.05, and a power of 90% was targeted. The effect size, assessed by Cohen's d, was predicted based on a mean difference of 2 points in the Visual Analog Scale (VAS), as it is commonly considered the minimum clinically relevant change in many chronic pain conditions [55], and the pooled standard deviation (SD_pooled). This was derived from the standard deviations of the intervention and control groups in a preliminary study, which were 1.29 and 2.61, respectively.

In recognition of the potential underestimation of standard deviations in the pilot study, the original SD values were adjusted using the Upper Confidence Limit (UCL) method, as described by Whitehead et al. [56], based on Browne's [57] approach. This adjustment, accounting for a 95% confidence level, yielded an SD of 3.227 using the formula [56, 57]: $s\_UCL^2 = [k/(X\_(1-X,k)^2)] s^2$.

Consequently, the effect size (d) was calculated as 0.619195 (derived from 2/3.23). With a 1:1 allocation ratio between groups, G*Power [58] indicated a requisite minimum of 56 subjects per group to detect a 2-point difference in VAS with 90% power. Considering a 15% dropout rate, the final sample size was set at 66 subjects per group, totaling 132 subjects for the study to maintain the desired statistical power.

## Recruitment

**Allocation.** *Sequence generation, concealment mechanism, and implementation.* Individuals meeting the inclusion criteria will be randomly allocated to one of two treatment arms: (1) POBTE plus exercise or (2) exercise therapy alone. Randomization will be conducted using computer-generated random numbers in block sizes ranging from 8 to 12, accommodating the requirements of the group intervention and ensuring that each group consists of 4 to 6 participants for the intervention to be feasible. The allocation will be concealed in an opaque, sealed envelope. A research assistant will open these envelopes and assign patients to their respective treatment arms. Multiple randomizations will be performed corresponding to the number of intervention-control groups formed. The randomization process will take place after obtaining informed consent and conducting baseline assessments.

## Blinding

**Masking.** In this trial, both outcome assessors and data analysts will be blinded after intervention assignment. They will not know the participants' intervention group affiliations during outcome assessment or data analysis. Due to the intervention's nature, involving

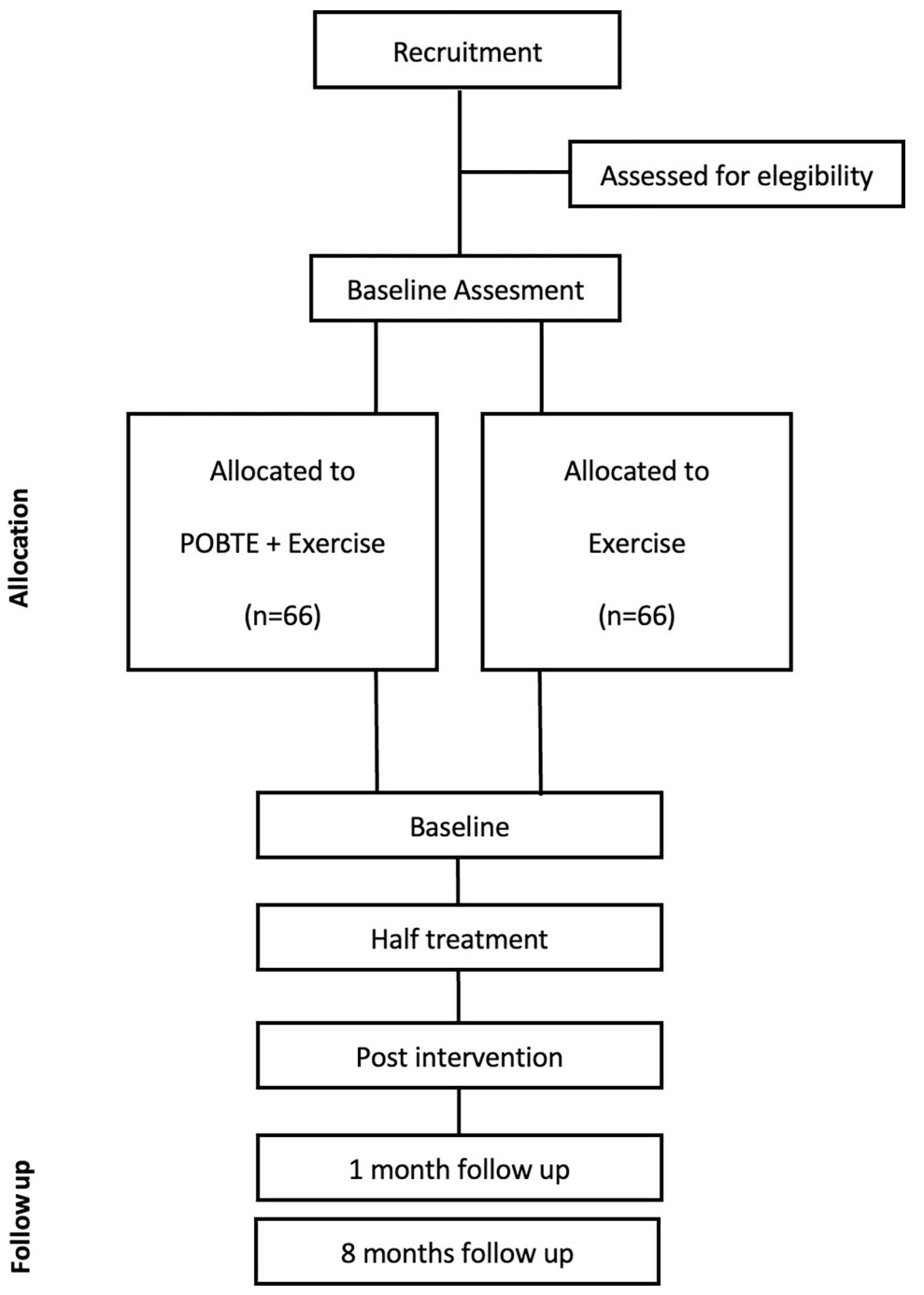

**Fig 3. Flow diagram.**

educational sessions and exercises requiring interaction, participants and educators cannot be blinded.

**Emergency unmasking.** A secure process, such as a password-protected system, will be employed for the emergency unmasking. Unbinding should only occur when deemed necessary, with the objective of preserving the trial's integrity.

**Data collection plan.** The trial will collect baseline and outcome data through questionnaires, laboratory tests, and physical assessments, using standardized and validated

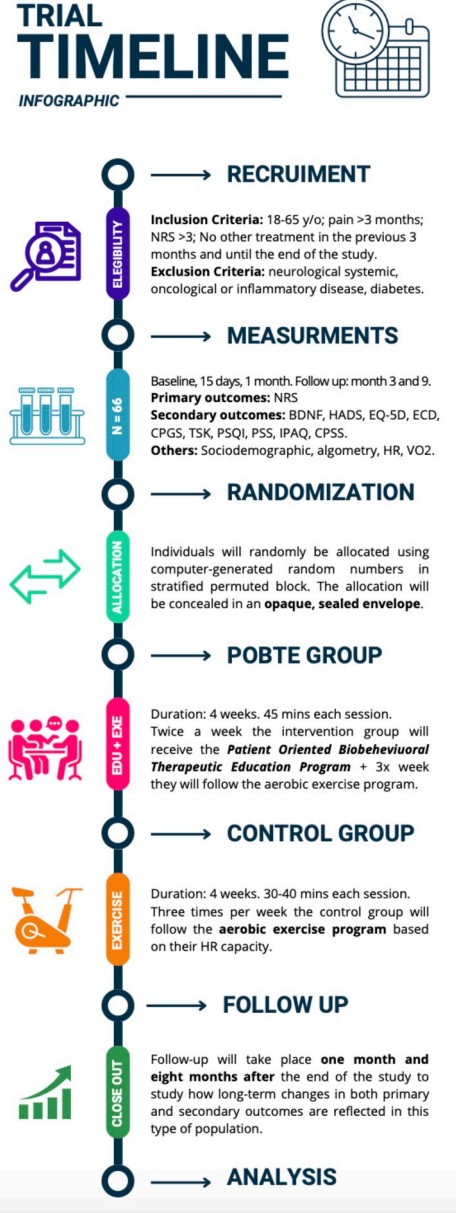

**Fig 4. Trial timeline.**

instruments to ensure data quality. The instruments used in the trial are validated, and information on them can be found in the protocol.

**Data collection plan: Retention.** To ensure that participants remain engaged and complete the follow-up procedure in our trial, several strategies have been implemented. Firstly, clear and concise information is provided to participants, explaining the trial's details, potential benefits, and associated risks, to help them understand the importance of their participation. Secondly, clear communication channels were established between the research team and participants to encourage an open and trusting relationship. The environment encourages participants to ask questions and provide feedback comfortably. Additionally, regular follow-up

| | STUDY PERIOD | | | | | | |
|---|---|---|---|---|---|---|---|
| | Enrolment | Allocation | Post-allocation | | | Close-out | |
| Timepoint | t-1 | 0 | t-baseline | t1 (15 days) | t-postint | t3 (1 month later) | t4 (8 months later) |
| ENROLMENT: | | | | | | | |
| Elegibility screen | X | | | | | | |
| Informed consent | X | | | | | | |
| Allocation | | X | | | | | |
| INTERVENTIONS | | | | | | | |
| [POBTE + Exercise] | | | ←——————————→ | | | | |
| [Exercise] | | | ←——————————→ | | | | |
| ASSESMENTS | | | | | | | |
| Visual Analogue Scale | | | X | X | X | X | X |
| Pressure Pain Thresholds | | | X | X | X | X | X |
| Brain Derived Neurotrophic Factor | | | X | X | X | X | X |
| Hospital Anxiety and Depression Scale | | | X | X | X | X | X |
| Pain catastrophizing scale | | | X | X | X | X | X |
| Quality of Life | | | X | X | X | X | X |
| Chronic Pain Grading Scale | | | X | X | X | X | X |
| Tampa Scale for Kinesiophobia | | | X | X | X | X | X |
| Pittsburgh Sleep Quality Index | | | X | X | X | X | X |
| Perceived Stress Scale | | | X | X | X | X | X |
| International Physical Activity Questionnaire | | | X | X | X | X | X |
| Knowledge questionnaire | | | X | X | X | X | X |
| Global Perception of Change | | | X | X | X | X | X |
| Height and Weight | | | X | X | X | X | X |
| Sociodemographic outcomes | | | X | X | X | X | X |
| Blood pressure | | | X | X | X | X | X |
| Respiratory rate | | | X | X | X | X | X |
| Saturation | | | X | X | X | X | X |
| HR | | | X | X | X | X | X |
| GROC Scale | | | | | | | X |

**Fig 5. SPIRIT figure showing the schedule of enrolment, interventions, and assessments.**

phone calls or emails are conducted to remind participants of upcoming study visits and to address any concerns they may have.

**Data management.** All data will be entered into a secure database system with restricted access. Data quality will be maintained through the implementation of range checks and double data entry processes. Tight data security and storage plans are in place, including regular backups, restricted access, and secure transfer methods.

**Statistics: Outcome.** Data analysis will be conducted using the statistical software SPSS version 25.0 (SPSS Inc., Chicago, IL, USA). A descriptive analysis of the demographic characteristics and pain intensity of the sample size will be performed, presenting continuous variables as mean ± standard deviation (SD) and 95% confidence interval (CI), while categorical variables will be presented as number (n) and percentage (relative frequency, %). In instances where quantitative variables follow a non-normal distribution, they will be described using the median and interquartile range.

For the comparison between groups, parametric tests based on the central limit theorem will be chosen, since the sample size of both groups will be greater than 30 [56]. A mixed-model Analysis of Covariance (ANCOVA) will be used including baseline as the covariate, to obtain between-group adjusted mean differences at each follow-up point, controlling for type I error rate using Bonferroni's correction. The chi-square test will be used for nominal variables. If the criteria for parametric testing cannot be met, robust methods will be applied [57]. A p-value $< 0.05$ will be accepted as statistically significant.

**Statistics: Additional analyses.** We will employ a multiple linear regression model to evaluate the strength of the associations between multiple predictor variables and the outcome variable at baseline. By using this method, we aim to discern the relative contribution of each predictor to the outcome, controlling for the potential influence of other variables, which will

provide a comprehensive understanding of the factors that are most significantly associated with the baseline measures. We have calculated the sample size required to ensure a minimum power for each of the individual regression coefficients of the model. The sample size calculation was based on Hsieh et al. [59]. It was expected a minimum correlation coefficient of 0.6 between the predictors and the dependent variable, and an interpreter correlation of 0.5, along with a total of 8 predictors and a statistical significance level of 0.05, we determined that a minimum of 53 subjects is needed to achieve a desired power of 90% (beta of 0.1), assuming a loss rate of 15%. Thus, we ensured the feasibility of these tests with the previously reported sample size calculation for the main outcome.

**Statistics: Analyzes population and missing data.** The analysis will be based on the intention-to-treat principle, including all randomized participants in the analysis, regardless of their compliance with the intervention protocol. Missing data will be addressed using appropriate imputation methods, such as multiple imputations or the last observation carried forward. Sensitivity analyses will be conducted to evaluate the impact of different imputation methods on the results. The analysis population will be clearly defined, and any deviations from the intention-to-treat principle will be reported and justified.

**Data monitoring: Formal committee.** This small-scale, low-risk intervention trial does not require a Data Monitoring Committee (DMC). Instead, the study investigators will take on the responsibility of monitoring trial data. The primary investigator will regularly review study data to ensure protocol adherence and to maintain the safety and well-being of participants. In the event of significant safety concerns or other issues, the primary investigator will collaborate with the study team and, if deemed necessary, seek guidance from an independent expert to determine the best course of action. Any concerns or protocol deviations will be promptly reported to the relevant regulatory authorities.

**Data monitoring: Interim analysis.** Although data and blood samples will be collected for 15 days during the intervention period, no interim analyses are planned for this trial. Nevertheless, it could be of interest to check potential changes in BDNF levels, which serve as a biomarker of brain plasticity, in relation to the learning curve [60] observed in patients. It is expected that the learning process during the intervention will lead to significant changes in BDNF levels.

**Harms.** The research team will vigilantly monitor participants for adverse events related to the intervention or trial conduct. Solicited events will be assessed during follow-up visits, while spontaneously reported events will be identified during unscheduled interactions.

Adverse events will be documented, and the principal investigator will conduct a thorough review to determine whether they require reporting to the ethics committee and regulatory authorities. Unintended effects coming from the trial interventions or conduct will also be closely monitored and appropriately managed. All events will be reported in the trial report, publication, and to the ethical committee.

**Research ethics approval.** To get approval for our study from a research ethics committee (REC), we developed a detailed protocol, submitted the requisite documentation, addressed any queries or concerns raised by the REC, obtained REC approval, and will ensure ongoing compliance. Additionally, we have established plans to communicate important protocol modifications to investigators, the REC, trial participants, trial registries, journals, and regulators.

**Consent or assent.** Informed consent or assent will be obtained from potential trial participants or authorized surrogates by trained research staff. The consent process will involve the use of written informed consent documents, which will be provided to the potential participants or their authorized surrogates in advance of the trial. These documents will include a detailed description of the study, the associated risks and benefits of participation, as well as the rights and responsibilities of the participants. Research staff will also be readily available to answer any questions or concerns that may arise during the consent process.

**Consent or assent: Ancillary studies.** We have included provisions in the informed consent form to cover potential ancillary studies involving participant data and biological specimens. The form clearly states that data and specimens may be used for future research with ethical approval, and participants have the right to withdraw their consent at any time. This approach ensures that participants are fully informed and that their rights and privacy are protected.

**Confidentiality.** Throughout the trial, we will protect participants' personal information. During the screening process, only essential information will be collected, and any unused data will be destroyed. Personal information will be kept secure and restricted to authorized members of the study team, identified solely by participant ID numbers. Upon completion of the trial, identifying information will be removed, and data will be securely stored for a specified period. Participants will be informed of our confidentiality policies during the informed consent process.

**Data access.** No datasets were generated or analysed during the current study. All relevant data from the RCT will be made available upon study completion.

**Ancillary and post-trial care.** The study team will develop a plan to provide ancillary or post-trial care for any harm experienced by participants during a clinical trial. Participants will be informed about these plans during the informed consent process and will work with regulatory authorities and review boards to ensure the implementation of appropriate provisions.

**Dissemination policy: Trial results.** We will communicate the trial results to participants, healthcare professionals, and the public. Results will be published in a peer-reviewed journal, presented at conferences, and reported in publicly accessible databases. Lay summaries will be provided to participants, and healthcare professionals will receive information. We will publish results regardless of the outcome, share the protocol, dataset, and statistical analysis, and adhere to responsible research guidelines. Our communication will be clear, timely, and accurate, with integrity and transparency.

**Dissemination policy: Authorship.** Authorship eligibility guidelines will follow international standards [61]. The contributions of eligible individuals will be clearly stated in publications, and professional writers will only be acknowledged if they meet the authorship eligibility guidelines.

**Dissemination policy: Reproducible research.** After the study, we will share the complete protocol, de-identified participant-level dataset, and statistical analysis, encouraging transparency and enabling further research. To enhance the accessibility and reproducibility of study findings, we will provide technical support and educational resources.

## Discussion

The current randomized controlled trial (RCT) seeks to examine the impact of a specific therapeutic education program, combined with exercise, on BDNF levels and perceived pain intensity among individuals suffering from chronic musculoskeletal pain. This article not only seeks to offer a more rigorous approach to therapeutic education from a biobehavioral perspective but also assesses the outcomes of an intervention that encompasses both the cognitive aspects and pain-related plasticity biomarkers, such as BDNF. Consequently, it represents a valuable contribution to low-cost, non-pharmacological treatments in physiotherapy. Although the study design and methods have been meticulously planned, several practical and operational considerations need to be addressed.

One potential limitation of our study pertains to the heterogeneity of chronic pain. Despite having well-defined inclusion criteria, it is important to acknowledge that chronic pain is a complex and multifactorial condition [62–64]. While we have not imposed restrictions on the specific site of pain in our selection criteria, a stance supported by evidence suggesting that

chronic pain is centrally mediated rather than solely dependent on the location of the injury [65–68], it is evident that various individual factors influence the pain experience in each patient [69–71]. These factors may impact the results obtained. To address this limitation, in addition to the inclusion criteria, we will consider conducting subgroup analysis if deemed necessary and feasible. This analysis will enable us to examine possible clusters of patients with similar characteristics, thereby enhancing our understanding of differences and treatment responses within our sample. It is also important to highlight that, regarding the proposed statistical analysis, it is assumed that categorical variables with more than five categories can be considered as continuous variables. This approach has been extensively considered in scientific literature and previously validated [72]. However, this could influence the final choice of the analysis method selected for the study.

Another concern is the potential confounding effects of medications and other interventions that participants might be undergoing. To address this issue, participants will be strongly advised to maintain their regular medication regimen (if applicable) but will not be permitted to initiate any new treatments during the program and follow-up period. This measure is implemented to minimize potential confounding factors and ensure the integrity of the study results. The study team will diligently screen and monitor participants to ensure compliance with this requirement. Additionally, the statistical analysis will consider adjusting for any emerging confounding variables.

Finally, it is important to note that the measurement of BDNF levels can pose challenges due to the inherent variability in sample collection, storage, and analysis. In this study, BDNF levels will be specifically measured in plasma samples, as they have been found to be more stable during analysis compared to serum samples [73]. To ensure precise and reliable results, the study team will adhere to rigorous protocols for plasma sample collection, processing, and storage. The analysis will be conducted using standardized ELISA kits, with regular calibration and quality control checks implemented to minimize potential sources of variation. These measures are crucial for ensuring the accuracy and validity of the BDNF measurements and upholding the integrity and reliability of the study's findings.

In conclusion, this randomized controlled trial serves as a significant milestone in two important aspects. Firstly, it contributes to the development of a more structured and rigorous therapeutic education model for individuals with chronic musculoskeletal pain by implementing validated checklists and protocols [31, 32]. This standardized approach ensures the delivery of effective interventions tailored to the specific needs of these patients.

Secondly, the study investigates the changes occurring at the central nervous system level by measuring biomarkers such as BDNF. This provides valuable insights into the underlying mechanisms of chronic pain and the impact of non-pharmacological interventions on brain plasticity. By elucidating the relationship between therapeutic education, BDNF levels, and pain outcomes, this research can pave the way for novel and targeted strategies to enhance treatment efficacy and improve the quality of life for individuals suffering from chronic pain.

## Supporting information

**S1 Checklist. TiDier checklist.**
(DOCX)

**S2 Checklist. CERT checklist.**
(DOCX)

**S3 Checklist. GREET checklist.**
(DOCX)

**S4 Checklist. SPIRIT checklist for this protocol.**
(DOC)

**S1 File. POBTE explanation.**
(DOCX)

**S2 File. Informed consent in Spanish.**
(DOCX)

**S3 File. Informed consent in English.**
(DOCX)

## Acknowledgments

Special acknowledgment is extended to the Spanish Ministry of Education's *FPU program* (FPU20/00041) for supporting the predoctoral contract of Silvia Di Bonaventura. This support has significantly contributed to the progress of this research. It is important to note that the program has not provided direct funding for the study itself.

### Informed consent materials

The model consent form and related documentation will be appended to the supporting information section of our study (S2 and S3 Files), ensuring a clear record of the consent process that can be easily accessed and verified by future reviewers.

### Biological specimens

We will collect and label blood samples, isolate plasma, and measure BDNF levels using an ELISA kit. Quality control measures will be implemented to ensure accurate measurements. Specimens will be stored in secure, monitored freezers (-80˚) at the research site and laboratory and tracked using unique participant identifiers. With proper ethics approval and participant consent, stored specimens may be used in future ancillary studies. This plan ensures the proper handling and availability of biological specimens for future research.

## Author Contributions

**Conceptualization:** Silvia Di Bonaventura, Raúl Ferrer-Peña.

**Formal analysis:** Silvia Di Bonaventura.

**Funding acquisition:** Raúl Ferrer-Peña.

**Investigation:** Silvia Di Bonaventura.

**Methodology:** Raúl Ferrer-Peña.

**Project administration:** Silvia Di Bonaventura.

**Supervision:** Raúl Ferrer-Peña.

**Validation:** Raúl Ferrer-Peña.

**Visualization:** Silvia Di Bonaventura.

**Writing – original draft:** Silvia Di Bonaventura.

**Writing – review & editing:** Josué Fernández Carnero, Raúl Ferrer-Peña.

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
