## [Decision Letter · Decision Letter 0]

7 Nov 2023

PONE-D-23-22066Can a specific biobehavioral based therapeutic education program lead to changes in pain perception and brain plasticity biomarkers in chronic pain patients? A study protocol for a randomized clinical trial.

PLOS ONE

Dear Dr. Di Bonaventura,

Thank you for submitting your manuscript to PLOS ONE. After careful consideration by a Reviewer and an Academic Editor, all of the critiques by the Reviewer must be addressed in detail in a revision to determine publication status. If you are prepared to undertake the work required, I would be pleased to reconsider my decision, but revision of the original submission without directly addressing the critiques of the Reviewer does not guarantee acceptance for publication in PLOS ONE. If the authors do not feel that the queries can be addressed, please consider submitting to another publication medium. A revised submission will be sent out for re-review. The authors are urged to have the manuscript given a hard copyedit for syntax and grammar.

We look forward to receiving your revised manuscript.

Kind regards,

Stephen D. Ginsberg, Ph.D.

Section Editor

PLOS ONE

“This study is part of PhD thesis of Silvia Di Bonaventura and will be supported by Spanish Ministry of Education (Grant number: FPU20/00041).”

“The authors received financial support for the future randomized clinical trial from the CSEU La SALLE (Universidad Autónoma de Madrid) with grant number: 2022A36005 but not for this protocol.

The authors who have received funding are not part of the present study as it is a shared project.”

4. We notice that your supplementary figures (Fig S1-S5)  are included in the manuscript file. Please remove them and upload them with the file type 'Supporting Information'. Please ensure that each Supporting Information file has a legend listed in the manuscript after the references list.

5. We note that the original protocol that you have uploaded as a Supporting Information file contains an institutional logo. As this logo is likely copyrighted, we ask that you please remove it from this file and upload an updated version upon resubmission.

**Comments to the Author**

1. Does the manuscript provide a valid rationale for the proposed study, with clearly identified and justified research questions?

Reviewer #1: Yes

2. Is the protocol technically sound and planned in a manner that will lead to a meaningful outcome and allow testing the stated hypotheses?

Reviewer #1: Partly

3. Is the methodology feasible and described in sufficient detail to allow the work to be replicable?

Reviewer #1: Yes

4. Have the authors described where all data underlying the findings will be made available when the study is complete?

Reviewer #1: Yes

5. Is the manuscript presented in an intelligible fashion and written in standard English?

Reviewer #1: No

6. Review Comments to the Author

You may also provide optional suggestions and comments to authors that they might find helpful in planning their study.

Reviewer #1: Important note: This review pertains only to ‘statistical aspects’ of the study and so ‘clinical aspects’ [like medical importance, relevance of the study, ‘clinical significance and implication(s)’ of the whole study, etc.] are to be evaluated [should be assessed] separately/independently. Further please note that any ‘statistical review’ is generally done under the assumption that (such) study specific methodological [as well as execution] issues are perfectly taken care of by the investigator(s). This review is not an exception to that and so does not cover clinical aspects {however, seldom comments are made only if those issues are intimately / scientifically related & intermingle with ‘statistical aspects’ of the study}. Agreed that ‘statistical methods’ are used as just tools here, however, they are vital part of methodology [and so should be given due importance]. I look at the manuscript in/with statistical view point, other reviewer(s) look(s) at it with different angle so that in totality the review is very comprehensive. However, there should be efforts from authors side to improve (may be by taking clues from reviewer’s comments). Therefore, please do not limit the revision only (with respect) to comments made here.

COMMENTS: I note that this manuscript is well drafted [and the study protocol is alright with respect to few aspects], I have certain definite observations/concerns (different opinion) which are given below:

Firstly, the account given in section ‘Sample Size’ (lines 354-368) appears alright, however, mind you that the effect size of 0.94 assumed is very large [your calculations are correct but the assumption regarding effect size in not at all reasonable, in my opinion]. Whereas details given of “POBTE’s intervention” are appreciable, difference from “Active Control group (Exercise)” [as indicated/implied by effect size assumed] cannot be that large [generally for such studies on non-pharmacological interventions], I guess. Effect size assumed here while estimating required sample size is very large [most likely], by any standard and please note that the ‘effect size’ assumed should be reasonable/realistic, else the study is very likely ‘not to be able to’ detect a difference despite its presence. In this case ‘effect size’ can at best be ‘medium’. The quoted reference [56] by Cohen J. ‘Statistical power analysis for the behavioral sciences’ is excellent, however, it not properly used {used to estimate SDpooled, but data used were from some pilot study, as this reference [Cohen’s] never comment(s) on likely ‘effect size’}.

According to table-2 on page 158 of Jacob Cohen’s paper “A power primer” in Psychological Bulletin, 1992, vol.:112, pp 155-159 [which is a sort of summary of the above quoted excellent book by Cohen himself for medium effect size you need n=64 per group (type-I error=0.05, power=80%). Your answer is [lines 364-5] “resulting in the need for at least 25 subjects per group to be able to detect a 2-point difference in VAS with 90% power” and in line 367, you said “Consequently, the study will require a total of 64 subjects to reach the required statistical power”. Note that my question is not about software used [G*Power or reference 57] as this software is excellent. For the same assumption of drop-out rate {25% is in fact, which is much more than generally allowed in any standard clinical trial(s)}, the study will require a total of 160 subjects to reach even 80% power.

In the context of ‘Statistics: additional analyses’ section (lines 430 onwards), mind you that ‘Multiple linear regression models (this technique [in fact any regression technique(s) for that matter] is/are not originally developed for testing the ‘Group difference(s)’. Head-to-head comparison is expected, as this is an indirect/secondary/by-product testing, in my opinion). You said [lines 432-33] “Subgroup analyses will consider age, gender, and duration of pain” is very good but do you think that this sample size will suffice to do this? Moreover, note that though the measures/tools used are appropriate [examples: Pain intensity - Visual Analogue Scale (VAS); Plasma BDNF levels; HADS self-completion scale; EuroQoL-5D questionnaire (EQ-5D); Pain Catastrophism Scale (ECD); Chronic Pain Grading Scale (CPGS)], most of them are likely to yield data that are in ‘ordinal’ level of measurement [and not in ratio level of measurement for sure {as the score two times higher does not indicate presence of that parameter/phenomenon as double}]. Then application of suitable non-parametric (or distribution free) test(s) is/are indicated/advisable [even if distribution may be ‘Gaussian’ (also called ‘normal’)]. Agreed that there is/are no non-parametric test(s)/technique(s) available to be used as alternative in all situation(s), but should be used whenever/wherever they are available. Therefore, in short use suitable non-parametric test(s)/technique(s) while dealing with data that are in ‘ordinal’ level of measurement even if [despite that] the distribution may be ‘Gaussian’. Testing ‘normality’ in sample [by using any normality test(s)} is not required/desired while dealing with data that are in ‘ordinal’ level of measurement [as most of the normality tests are not valid for ‘ordinal’ data].

In ‘Conclusion’ section of abstract [lines 24-25] you conclude that “The rigorous scientific methods used ensure that the proposed interventions will be clinically applicable across different health care systems” is not understood. What type of conclusion it is? How this study helped the authors to conclude this? In any case, remember/mind you that this is a scientific/academic document and so all details should be clearly/correctly communicated (do not take readers’ for granted).

Refer to line 379 [stratified permuted block (block size of 4 and 6)], Please explain the role of ‘stratification’ in ‘permuted block’ randomization’? {i.e. how you used/executed in this study?} Moreover, please note that while using ‘permuted block randomization’, we generally do not reveal ‘block size’ [which is randomly chosen]. Further, please refer to lines 425-27 [ANOVA test will be used to analyze the group factor in the quantitative variables and if the ANOVA reveals statistical significance, a post-hoc analysis with Bonferroni correction will be performed], according to my understanding, you may not need ANOVA and subsequent any post-hoc analysis as there are only two groups.

In the beginning of the article (line 29) there is a list ‘Strengths and limitations of this study’ where there is hardly any limitation(s). Does that mean {according to authors} there are none?

As pointed out in ‘important note’ above “This review pertains only to ‘statistical aspects’ of the study and so ‘clinical aspects’ should be assessed separately/independently [one should carefully consider/look at the clinical implications of the study]. In my opinion, to make this article acceptable (which is not impossible), large amount of re-vision (re-drafting) may be needed. However, please do not limit the revision only (with respect) to comments made here [as this is a protocol and is possible now to accommodate suggestion by other respected reviewers. ‘Major revision’ is recommended.

7. PLOS authors have the option to publish the peer review history of their article (what does this mean?). If published, this will include your full peer review and any attached files.

**Do you want your identity to be public for this peer review?** For information about this choice, including consent withdrawal, please see our Privacy Policy.

Reviewer #1: No

---

## [Author Response · Author response to Decision Letter 0]

13 Nov 2023

Dear Journal Editors,

First and foremost, we would like to express our deepest gratitude for your detailed corrections and suggestions, which have been immensely helpful in enhancing our manuscript. Below, we detail the revisions made in response to each of your requirements:

Compliance with PLOS ONE Style Requirements and Figure Corrections: We have reviewed and adjusted our manuscript to ensure it meets the style requirements of PLOS ONE, including those related to file naming, using the provided templates. Additionally, we have corrected the figures using the Preflight Analysis and Conversion Engine (PACE) digital diagnostic tool to ensure they meet the required standards. Furthermore, the entire paper has been meticulously reviewed and edited by a specialized editor (PaperTrue) for English language and composition to ensure clarity and coherence.

Funding Information: We have made the necessary corrections in the 'Funding Information' sections to accurately reflect it. It is important to highlight that we have clearly differentiated two sources of funding:

a. Funding from CSEU La Salle: In the 'Funding Information' section, we have indicated that CSEU La Salle, with grant number: 2022A36005, will provide the necessary funds for acquiring the material for the study we are going to conduct.

b. Acknowledgment to the Spanish Ministry of Education: As per the requirement of the Spanish Ministry of Education, we should acknowledge the predoctoral contract (FPU20/00041) of one of the authors in the “Acknowledgments” section. However, this contract does not provide any direct financial contribution to the study. If it is not permissible to include this acknowledgment in that section due to the journal's policies, we kindly request guidance on where you would prefer it to be included and we will change it.

Supplementary Figures: We have removed the supplementary figures (Fig S1-S5) from the manuscript file and uploaded them as 'Supporting Information', ensuring that each has an appropriate legend listed after the references.

Removal of Institutional Logo from Project Report: Regarding the Supporting Information file containing the project report, we have removed the institutional logo from both the English version and the original Spanish version, to avoid copyright issues.

Protocol Registration in Pprotocols.io: Our protocol was initially registered in MedRxiv. Following your advice, we have now also registered it in protocols.io.

Additionally, we have addressed all the reviewer's comments in a separate document, ensuring a thorough and detailed response to each point raised.

We are fully willing and available to make any additional adjustments that may be necessary. We look forward to your comments and guidance to proceed towards the publication of the manuscript.

Responses to Reviewer’ comments:

Dear Reviewer,

Thank you for your insightful commentaries aimed at enhancing the quality of our manuscript. We have undertaken a comprehensive and detailed modification process in accordance with your valuable suggestions.

In the following lines, we will explain in detail, question by question, how we have addressed and modified the manuscript in response to each specific point you raised. 

REVIEWER

1. In response to “Firstly, the account given in section ‘Sample Size’ (lines 354-368) appears alright, however, mind you that the effect size of 0.94 assumed is very large [your calculations are correct but the assumption regarding effect size in not at all reasonable, in my opinion]. Whereas details given of “POBTE’s intervention” are appreciable, difference from “Active Control group (Exercise)” [as indicated/implied by effect size assumed] cannot be that large [generally for such studies on non-pharmacological interventions], I guess. Effect size assumed here while estimating required sample size is very large [most likely], by any standard and please note that the ‘effect size’ assumed should be reasonable/realistic, else the study is very likely ‘not to be able to’ detect a difference despite its presence. In this case ‘effect size’ can at best be ‘medium’.

We would like to express our sincere gratitude for your comments, which have resulted in a thorough revision of the initial assumptions of our manuscript regarding effect size. Considering them, we recognize that the initially proposed effect size of 0.94 was indeed ambitious for nonpharmacological interventions, despite being based on the results of our pilot study. Accordingly, we have recalibrated our calculations, adopting a "medium" effect size in line with the standard literature and better reflecting the expected results in such studies. You can find in line 419-441.

2. In response to “The quoted reference [56] by Cohen J. ‘Statistical power analysis for the behavioral sciences’ is excellent, however, it not properly used {used to estimate SDpooled, but data used were from some pilot study, as this reference [Cohen’s] never comment(s) on likely ‘effect size’}. According to table-2 on page 158 of Jacob Cohen’s paper “A power primer” in Psychological Bulletin, 1992, vol.:112, pp 155-159 [which is a sort of summary of the above quoted excellent book by Cohen himself for medium effect size you need n=64 per group (type-I error=0.05, power=80%). Your answer is [lines 364-5] “resulting in the need for at least 25 subjects per group to be able to detect a 2-point difference in VAS with 90% power” and in line 367, you said “Consequently, the study will require a total of 64 subjects to reach the required statistical power”. Note that my question is not about software used [GPower or reference 57] as this software is excellent. For the same assumption of drop-out rate {25% is in fact, which is much more than generally allowed in any standard clinical trial(s)}, the study will require a total of 160 subjects to reach even 80% power”:

According to the calculations adjusted for the Standard Deviations previously mentioned, the effect size (d) obtained was 0.619195, based on that and performing again the calculation in G*Power to maintain the desired power of 90% we obtained a total of 56 subjects per group, very close to the recommendation made in your commentary. In addition, we have incorporated a 15% loss rate in our sample size calculation (132 total subjects, rounding up to the next even number), more in line with what is to be expected in this type of trial. According to the Cohen cite in our previous version of the manuscript, we have taken steps to clarify in our manuscript the references used for the calculation of the pooled standard deviation that were derived from our pilot study data. We believe that these modifications significantly improve the methodological rigor and clarity of our study, and we appreciate the opportunity to improve our work based on your valuable input. (Line 436-441).

3. In response to “In the context of ‘Statistics: additional analyses’ section (lines 430 onwards), mind you that ‘Multiple linear regression models (this technique [in fact any regression technique(s) for that matter] is/are not originally developed for testing the ‘Group difference(s)’. Head-to-head comparison is expected, as this is an indirect/secondary/by-product testing, in my opinion). You said [lines 432-33] “Subgroup analyses will consider age, gender, and duration of pain” is very good but do you think that this sample size will suffice to do this? Moreover, note that though the measures/tools used are appropriate [examples: Pain intensity - Visual Analogue Scale (VAS); Plasma BDNF levels; HADS self-completion scale; EuroQoL-5D questionnaire (EQ-5D); Pain Catastrophism Scale (ECD); Chronic Pain Grading Scale (CPGS)], most of them are likely to yield data that are in ‘ordinal’ level of measurement [and not in ratio level of measurement for sure {as the score two times higher does not indicate presence of that parameter/phenomenon as double}]. 

We have carefully reviewed your comment and recognize that these models were not primarily intended to examine differences between groups, so we have adjusted our manuscript to accurately reflect this and avoid misunderstandings. This adjustment corresponds to one of our secondary objectives, which is to investigate the relationship between the main variable and the secondary variables of this study (baseline levels). Regarding concerns about sample size for subgroup analyzes that include age, gender, and pain duration, after reevaluating the study sample size and statistical power, we consider that, if the necessary conditions are met, this type of analysis can be carried out with the variables provided in the study. (Line 522-531)

4. In response to “Then application of suitable non-parametric (or distribution free) test(s) is/are indicated/advisable [even if distribution may be ‘Gaussian’ (also called ‘normal’)]. Agreed that there is/are no non-parametric test(s)/technique(s) available to be used as alternative in all situation(s), but should be used whenever/wherever they are available. Therefore, in short use suitable non-parametric test(s)/technique(s) while dealing with data that are in ‘ordinal’ level of measurement even if [despite that] the distribution may be ‘Gaussian’. Testing ‘normality’ in sample [by using any normality test(s)} is not required/desired while dealing with data that are in ‘ordinal’ level of measurement [as most of the normality tests are not valid for ‘ordinal’ data]. 

Thank you so much for the appreciation regarding the use of categorical variables and ordinal level measurements in our tools, such as the Visual Analogue Scale and the others mentioned, we have considered the guidelines of relevant studies, as Rhemtulla et al. (2012), who suggest that under certain conditions, categorical variables with more than five categories can be treated as continuous. However, we are prepared to employ robust methods if the conditions for parametric testing are not met, thus ensuring the validity of our statistical analysis. We have added this explanation in the manuscript more precisely. (Line 505-513)

5. In response to “Refer to line 379 [stratified permuted block (block size of 4 and 6)], Please explain the role of ‘stratification’ in ‘permuted block’ randomization’? {i.e. how you used/executed in this study?} Moreover, please note that while using ‘permuted block randomization’, we generally do not reveal ‘block size’ [which is randomly chosen]. 

After careful consideration of your feedback, we have made significant revisions to that approach. In our initial submission, we incorrectly mentioned 'stratification' in the context of our 'permuted block' randomization. This was a misinterpretation on our part. Stratification was not utilized in our study; therefore, we have removed this term to prevent any misunderstanding about our randomization process.

Additionally, in line with the standard practices of 'permuted block randomization', we have revised our manuscript to exclude the explicit mention of block sizes. We acknowledge that disclosing the block sizes could potentially introduce bias into the study. The randomization will now be conducted using computer-generated random numbers with undisclosed block sizes to ensure unpredictability and methodological soundness. (Line 448-451)

Further, please refer to lines 425-27 [ANOVA test will be used to analyze the group factor in the quantitative variables and if the ANOVA reveals statistical significance, a post-hoc analysis with Bonferroni correction will be performed], according to my understanding, you may not need ANOVA and subsequent any post-hoc analysis as there are only two groups.

Furthermore, we have also realized, thanks to your observation, that our initial manuscript did not adequately explain our analytical approach regarding the interaction between time and group factors. We will indeed be employing an ANCOVA for this purpose. The manuscript has now been amended to include a thorough explanation of how we intend to explore this interaction, providing a clear rationale for the use of ANCOVA in this context. (Line 505-513 / Line 522-531)

6. In response to “In ‘Conclusion’ section of abstract [lines 24-25] you conclude that “The rigorous scientific methods used ensure that the proposed interventions will be clinically applicable across different health care systems” is not understood. What type of conclusion it is? How this study helped the authors to conclude this? In any case, remember/mind you that this is a scientific/academic document and so all details should be clearly/correctly communicated (do not take readers’ for granted).

Thank you for your comment regarding the conclusion in our abstract. We understand your concern about the clarity of our statement. The intent was to express that the methods we have employed are robust and, as such, the findings could be relevant to various health care settings. However, we acknowledge that this may have been an overstatement. To this end, we have placed greater emphasis on explaining the impact that the study may have in terms of reproducibility, providing a more comprehensive treatment approach, and objectifying the results (line 42-46).

7. In response to “In the beginning of the article (line 29) there is a list ‘Strengths and limitations of this study’ where there is hardly any limitation(s). Does that mean {according to authors} there are none?

We really appreciate you pointing out the need to address potential limitations in our study. Upon reviewing line 29, we recognize that the original manuscript did not adequately detail the main limitations. To rectify this, we have revised the section 'Strengths and limitations of this study” and included different limitations. More specifically, two principal factors that could substantially impact our findings. Firstly (line 76-71), the instability of BDNF as a biomarker is a critical concern, given its sensitivity to minor changes such as dietary variations, levels of physical activity, stress, and even ambient temperature conditions. Such fluctuations necessitate cautious interpretation of BDNF levels in clinical research settings. Secondly (line 73-77), the patient demographic for studies of this nature, for instance, those with fibromyalgia or migraine conditions, is highly specific and not broadly generalizable. This specificity is pivotal in understanding the context and applicability of our results. These primary factors, along with secondary considerations mentioned in the discussion, are crucial as they have been shown to influence outcomes in prior research significantly.

8. Is the manuscript presented in an intelligible fashion and written in standard English? PLOS ONE does not copyedit accepted manuscripts, so the language in submitted articles must be clear, correct, and unambiguous. Any typographical or grammatical errors should be corrected at revision, so please note any specific errors here. Reviewer #1: No.

We acknowledge and appreciate your recommendation to have our manuscript reviewed by a language expert, considering English is not our first language. We have since engaged a professional editor, whose expertise has greatly enhanced the clarity and readability of our text. A certificate of the editing will be provided as an attachment to confirm this revision.

Once again, thanks for the possibility to review this manuscript. We deeply appreciate the time and effort you have invested in reviewing our manuscript and providing valuable feedback. We are fully prepared and willing to make any further changes if necessary, to ensure that our manuscript meets the standards of the journal.

Sincerely,

The authors.

---

## [Decision Letter · Decision Letter 1]

27 Nov 2023

PONE-D-23-22066R1Can a specific biobehavioral-based therapeutic education program lead to changes in pain perception and brain plasticity biomarkers in chronic pain patients? A study protocol for a randomized clinical trial.

PLOS ONE

Dear Dr. Di Bonaventura,

Thank you for resubmitting your work to PLOS ONE. Please make the corrections posed by Reviewer #1 so I can render a decision on this manuscript.

We look forward to receiving your revised manuscript.

Kind regards,

Stephen D. Ginsberg, Ph.D.

Section Editor

PLOS ONE

Journal Requirements:

**Comments to the Author**

1. Does the manuscript provide a valid rationale for the proposed study, with clearly identified and justified research questions?

Reviewer #1: Partly

2. Is the protocol technically sound and planned in a manner that will lead to a meaningful outcome and allow testing the stated hypotheses?

Reviewer #1: No

3. Is the methodology feasible and described in sufficient detail to allow the work to be replicable?

Reviewer #1: No

4. Have the authors described where all data underlying the findings will be made available when the study is complete?

Reviewer #1: Yes

5. Is the manuscript presented in an intelligible fashion and written in standard English?

Reviewer #1: Yes

6. Review Comments to the Author

You may also provide optional suggestions and comments to authors that they might find helpful in planning their study.

Reviewer #1: COMMENTS: Although all the comments are answered and few positively attended, frankly speaking I do not agree with Rhemtulla et al. (Psychol Methods, 2012 Sep;17(3):354-73) who suggested that under certain conditions, categorical variables with more than five categories can be treated as continuous [quoted as part of answer to my comment]. Look at few of their statements like “In our initial submission, we incorrectly mentioned 'stratification' in the context of our 'permuted block' randomization. This was a misinterpretation on our part.” Which hopefully throw light on quality of work.

7. PLOS authors have the option to publish the peer review history of their article (what does this mean?). If published, this will include your full peer review and any attached files.

**Do you want your identity to be public for this peer review?** For information about this choice, including consent withdrawal, please see our Privacy Policy.

Reviewer #1: No

---

## [Author Response · Author response to Decision Letter 1]

27 Nov 2023

Dear reviewer, thank you again for your comments.

REVIEWER

1.- In response to “Although all the comments are answered and few positively attended, frankly speaking I do not agree with Rhemtulla et al. (Psychol Methods, 2012 Sep;17(3):354-73) who suggested that under certain conditions, categorical variables with more than five categories can be treated as continuous [quoted as part of answer to my comment].”

We understand the reviewer's discrepancy, but we would like to emphasize that the method used in our study, based on the aforementioned study by Rhemtulla et al 2012, has been widely cited and accepted by the scientific community, specifically it has been used in more than 395 articles, including 14 in Q1 journals and 88 in Q2 of the JCR, and has even been used and referenced in previous studies published in prestigious journals, including PLOS ONE, as in the article with PMID 37603578 or the one with PMID 37310969, for example. We consider that such a method is sufficiently validated and explained, and that it is relevant to our study, although we acknowledge that it could influence future results in some way, based on the concerns of reviewer 1. This point has been included in the potential limitations of our study in the manuscript.

 

2.- In response to:

“Look at few of their statements like “In our initial submission, we incorrectly mentioned 'stratification' in the context of our 'permuted block' randomization. This was a misinterpretation on our part.” Which hopefully throw light on quality of work.”

We also consider that the methodological quality of our work is well documented and we believe that we have correctly responded to all the considerations set out in detail in the previous review, having considerably improved the initial manuscript, and we believe that our study meets the necessary quality requirements to be considered for publication, we regret if an error we made in the drafting of the first version of the manuscript, as English is not our native language, and which we have corrected and submitted for professional editing, is put forward as an example of the quality of the manuscript, but we can do nothing other than apologize for the error made initially and corrected subsequently. Nevertheless, we sincerely thank the reviewer 1 for the time spent in improving our manuscript.

---

## [Editor Report · Decision Letter 2]

18 Dec 2023

Can a specific biobehavioral-based therapeutic education program lead to changes in pain perception and brain plasticity biomarkers in chronic pain patients? A study protocol for a randomized clinical trial.

PONE-D-23-22066R2

Dear Dr. Di Bonaventura,

We’re pleased to inform you that your manuscript has been judged scientifically suitable for publication and will be formally accepted for publication once it meets all outstanding technical requirements.

Kind regards,

Stephen D. Ginsberg, Ph.D.

Section Editor

PLOS ONE

---

## [Editor Report · Acceptance letter]

9 Jan 2024

PONE-D-23-22066R2 

PLOS ONE

Dear Dr. Di Bonaventura, 

I'm pleased to inform you that your manuscript has been deemed suitable for publication in PLOS ONE. Congratulations! Your manuscript is now being handed over to our production team.

Kind regards, 

on behalf of

Dr. Stephen D. Ginsberg 

Section Editor

PLOS ONE